# Circulating CD4^+^ Treg, CD8^+^ Treg, and CD3^+^ γδ T Cell Subpopulations in Ovarian Cancer

**DOI:** 10.3390/medicina59020205

**Published:** 2023-01-20

**Authors:** Rong Li, Juan Xu, Ming Wu, Shuna Liu, Xin Fu, Wenwen Shang, Ting Wang, Xuemei Jia, Fang Wang

**Affiliations:** 1Department of Laboratory Medicine, The First Affiliated Hospital of Nanjing Medical University, Nanjing 210029, China; 2National Key Clinical Department of Laboratory Medicine, Nanjing 210029, China; 3Department of Gynecology, Women’s Hospital of Nanjing Medical University, Nanjing 210004, China; 4Department of Laboratory Medicine, The Affiliated Taizhou People’s Hospital of Nanjing Medical University, Taizhou 225300, China; 5Department of Clinical Laboratory, Children’s Hospital of Fudan University, National Children’s Medical Center, Shanghai 201102, China

**Keywords:** ovarian cancer, Tregs, subpopulations, immunologic surveillance, tumor markers

## Abstract

*Background and Objectives*: Regulatory T cells (Tregs) are usually enriched in ovarian cancer (OC), and their immunosuppressive function plays a key role in tumorigenesis and progression. We mainly explored the phenotypical characterization of Treg-related markers on αβ and γδ T cell subsets in patients with OC. *Materials and Methods*: Thirty-six untreated patients with OC at the Women’s Hospital of Nanjing Medical University from September 2019 to August 2021 were enrolled. Phenotypical characterization of Tregs-related markers were detected by flow cytometry (FCM). Enzyme-linked immunosorbent assay was used to detect the levels of carbohydrate antigen (CA125) and transforming growth factor β (TGF-β). The level of human epididymis protein 4 (HE4) was detected by electrochemiluminescence immunoassay. *Results*: Circulating CD4^+^ Tregs, CD8^+^ Tregs, and CD3^+^γδ T cell subpopulations from OC patients have elevated Foxp3, CD25, CD122, Vδ1, and reduced CD28 expression compared to benign ovarian tumor (BOT) patients and healthy controls (HC). The upregulation of Foxp3 and Vδ1 and the downregulation of CD28 were highly specific for maintaining the immunosuppression function of CD4^+^ Tregs, CD3^+^γδ T cells, and CD8^+^ Tregs in OC patients. These Treg subpopulations were able to discriminate OC from BOT and HC. The levels of CA125, HE4, and TGF-β were increased in OC patients. A significant positive correlation between Treg subpopulations and CA125, HE4, and TGF-β was revealed. *Conclusions*: Proportions of CD4^+^ Tregs, CD8+ Tregs, and CD3^+^γδ T cell subsets were significantly increased in OC patients and were positively correlated with FIGO stage/metastasis status, CA125, HE4, and TGF-β. These indicators have the potential to be used as immunosurveillance biomarkers for OC.

## 1. Introduction

OC is one of the most lethal gynecological malignancies. In the year 2020, there were approximately 21,400 new cases of carcinoma of the ovary, which was estimated to be 1.2% of all cases of cancer. The mortality related to it is 13,700. There is a 47.3% chance of survival for five years for women [1,2]. The use of modern diagnosis and treatment methods for ovarian cancer can reduce its mortality rate, although not enough data are available to compare different parts of the world in this regard [3]. Significant advances have been made in surgery management and systemic therapeutic approaches in OC. However, the majority of these patients will experience recurrence within 1 to 2 years following treatment [4]. The high lethality of a tumor is in part due to the infiltration of immunosuppressive cells into the tumor microenvironment (TME) [5,6], so a better understanding of the immunosuppressive potential in OC is critical.

Tregs belong to a lineage of immunosuppressive cells that function by suppressing effector T cells and immune-mediated inflammation [7]. Vanguri R et al. observed changes in immune cell subsets expressing repressive or stimulatory proteins resulting in immune compositions more favorable to checkpoint modulations, suggesting novel therapeutic strategies in the tumor recurrent setting [8]. Neoadjuvant chemotherapy was associated with increased densities of CD3^+^, CD8^+^, PD-1^+^, and CD20^+^ T cells, but other immune subsets and factors were unchanged [9]. CD4^+^ Tregs rapidly decreased after primary tumor debulking, whereas CD8^+^ CD25^+^ FOXP3^+^ Tregs are not detectable in peripheral blood. Similar results on CD4^+^ Tregs were observed with chemical debulking in women subjected to neoadjuvant chemotherapy [10].

Infiltration of Tregs in the TME is commonly associated with poor prognosis in various types of cancer, including OC [11,12]. It is now accepted that Tregs are heterogeneous in phenotype and function, with distinct subpopulations identified in human peripheral blood [13]. The assessment of specific functional subtypes of Tregs may be critical for a more accurate assessment of prognostic outcomes in OC [14,15]. In the past decade, researchers have witnessed an explosion in the studies on CD4^+^ Tregs, whereas research on another αβ Tregs type CD8^+^ Tregs, and γδ T cells that possess suppression function subsets have received considerably less attention.

In this study, we investigated whether circulating αβ (CD4^+^, CD8^+^) Tregs and γδ (CD3^+^Vδ1, CD3^+^Vδ2) T cell subpopulations from OC patients have unique marker characteristics and can be used as biomarkers to evaluate immunosuppressive potential, which has clinical significance. This study contributes to a better understanding of the heterogeneity of Tregs in OC TME and may provide new ideas for the identification of novel biomarkers in OC immunologic surveillance.

## 2. Materials and Methods

### 2.1. Patients and Specimens

We studied 36 untreated patients with OC at the Women’s Hospital of Nanjing Medical University from September 2019 to August 2021 (Table 1). At the same time, 32 BOT patients and 40 healthy volunteers were selected as controls [8,9].

All OC patients were staged according to the International Federation of Gynecology and Obstetrics. The inclusion criterias were listed as follows: (1) pathology confirmed diagnosis of OC; (2) none of these patients had received radiotherapy or chemotherapy before specimen collection; (3) without autoimmune diseases, severe renal and liver failure, incomplete pathological information, or severe underlying diseases.

### 2.2. Blood Processing and Flow Cytometry

Fasting venous blood samples were collected in ethylene diamine tetraacetie acid anticoagulated tubes after informed consent in different groups. Peripheral blood mononuclear cells (PBMC) were isolated by Lymphocyte Separation Medium (TBD, Tianjin, China). PBMC isolated from blood was re-suspended in 100 µL flow cytometry staining buffer.

Freshly isolated PBMCs were incubated with anti-CD3 (APC), anti-CD4 (FITC), anti-CD8 (FITC), anti-CD25 (APC), anti-CD28 (APC), and CD122 (BV421), Vδ2 (PE) (all from Biolegend, USA), and Vδ1 (FITC) (Abcam, Cambridge, UK) in 100 µL PBS for 20 min at room temperature in the dark. After 45 min of cell fixation and perforation of the nuclear envelope, cells were stained with anti-FoxP3 (PE) for 30 min. Finally, FACS Aria II (BD Biosciences, San Jose, CA, USA) was used to detect the fluorescence signal values, and FlowJo V10 software (FlowJo, Ashland, Wilmington, DE, USA) was used to analyze the proportion of each Tregs subset.

### 2.3. Enzyme-Linked Immunosorbent Assay (ELISA)

TGF-β levels in serum samples of OC patients (*n* = 36), BOT patients (*n* = 32), and HC (*n* = 40) were detected by the commercially available Human TGF-β ELISA Kit (eBioscience, Santiago, CA, USA).

### 2.4. Measurement of CA125, HE4 Concentrations

CA125 and HE4 values were obtained from patient records. The limits of normal values were 35 U/mL for CA125 and 140 pmol/L for HE4 in accordance with the clinical reference ranges used routinely at the Women’s Hospital of Nanjing Medical University.

### 2.5. Statistical Analysis

The statistical analysis was conducted by SPSS 22.0 (IBM) software (SPSS Inc, Chicago, IL, USA). The measurement data conforming to normal distribution were expressed as mean ± standard deviation (Means ± SDs), and non-normally distributed measurement data were expressed as median ((Interquartile range, M (P_25_, P_75_)). The differences between the two groups were compared using an independent samples *t*-test or the non-parametric Mann–Whitney U test. The association between variables and clinical characteristics was evaluated by Chi-square or Fisher exact test. *P <* 0.05 was considered statistically significant.

## 3. Results

### 3.1. Comparison of Proportion of Treg Subsets in Different Groups

We investigated the levels of CD4^+^ Tregs, CD8^+^ Tregs, and CD3^+^γδ T cell subsets in the circulation of OC patients, BOT patients, and HC (Figure 1, Table 2). We found that the proportion of Tregs (Foxp3^+^, CD25^+^Foxp3^+^) in CD4^+^ T cells was significantly higher in the peripheral blood samples of OC patients than in BOT patients and HC. We further confirmed that the presence of CD8^+^ Tregs subsets (CD8^+^CD28^−^, CD8^+^Foxp3^+^, CD8^+^CD28^−^Foxp3^+^, CD8^+^CD122^+^) was also significantly elevated in OC patients compared with BOT patients and HC. In addition, the proportion of CD3^+^Vδ1 T cells in OC patients was higher than that in BOT patients and HC (Figure 1, *P <* 0.05). A similar percentage of CD4^+^CD25^+^, CD4^+^Foxp3^+^, CD4^+^CD25^+^Foxp3^+^, CD8^+^CD28^−^, CD8^+^CD28^−^Foxp3^+^, CD8^+^CD122^+^ Tregs, and CD3^+^Vδ1 T cells were observed between BOT patients and HC (*P >* 0.05). We found no significant difference in CD3^+^Vδ2 T cells proportion among the three groups (*P >* 0.05).

We next investigated the ability of each Treg subpopulation to distinguish between OC patients, BOT patients, and HC (Figure 2). ROC analyses revealed that CD4^+^CD25^+^Foxp3^+^ Tregs (AUC:0.878 and 0.696, *P <* 0.0001 and *P <* 0.001), CD8^+^CD28^−^ Tregs (AUC:0.795 and 0.816, *P <* 0.0001), CD8^+^Foxp3^+^ Tregs [(AUC:0.908 and 0.737, *P <* 0.0001 and *P* = 0.0008), CD8^+^CD28^−^Foxp3^+^ Tregs [(AUC:0.878 and 0.776, *P <* 0.0001 and *P <* 0.0001), and CD3^+^Vδ1 T cells [(AUC:0.814 and 0.889, *P <* 0.0001 and *P <* 0.0001) could clearly both discriminate between OC patients and HC, and OC patients and BOT patients with *P <* 0.05. The remaining Treg subsets and CD3^+^Vδ2 T cells were poorly discriminative (*P <* 0.05). Altogether, our data suggest a potential role for circulating CD4^+^CD25^+^Foxp3^+^, CD8^+^CD28^−^, CD8^+^Foxp3^+^, and CD8^+^CD28^−^Foxp3^+^ Tregs and CD3^+^Vδ1 T cells in OC diagnosis.

### 3.2. The Relationship between Proportion of Different Treg Subsets and Clinical Characteristics of OC Patients

The above results show that Treg subset proportions in the peripheral blood of OC patients were related to FIGO stage, lymph node metastasis, distant metastasis, CA125, and HE4. The proportion of CD4^+^CD25^+^, CD4^+^Foxp3^+^, CD4^+^CD25^+^Foxp3^+^, and CD8^+^CD28^−^ Tregs in OC patients at stage III–IV was higher than that of patients at stage I–II (Table 3, *P <* 0.05; Table 4, *P <* 0.05). The proportion of CD4^+^CD25^+^Foxp3^+^, CD8^+^CD28^−^Treg in OC patients with lymph node metastasis was higher than that without lymph node metastasis according to the enhanced CT examination (Table 3, *P <* 0.05; Table 4, *P <* 0.05). Furthermore, OC patients with distant metastasis at diagnostic had higher CD4^+^CD25^+^, CD4^+^Foxp3^+^, CD4^+^CD25^+^Foxp3^+^, and CD8^+^CD28^−^ Tregs than OC patients without distant metastasis at diagnostic (Table 3, *P <* 0.05; Table 4, *P <* 0.05). Analysis showed that age, tumor size, and histology did not correlate with the Tregs proportion (Table 3, Table 4 and Table 5).

### 3.3. The Relationship between Proportion of Different Treg Subsets and CA125, HE4 of OC Patients

High levels of CA125 and HE4 status have been reported in OC patients [16,17]. We also investigated the levels of CA125 and HE4 in the serum samples of OC patients. In agreement with previous reports, we found that CA125 and HE4 in OC patients were elevated compared with BOT patients and HC (Figure 3a; *P <* 0.05).

High levels of CA125 and HE4 frequently correlate with poor prognosis [18]. Based on the strong correlation between Treg subsets and OC, we interrogated subpopulations of Tregs to identify which ones, if any, correlate with CA125 and HE4. The CD4^+^CD25^+^, CD8^+^CD28^−^, CD8^+^Foxp3^+^, CD8^+^CD28^−^Foxp3^+^, CD8^+^CD122^+^ Tregs, and CD3^+^Vδ1 T cells proportion in CA125 higher (>median) OC patients was higher than that in CA125 (<median) lower patients (Table 6; *P <* 0.05). The CD8^+^CD28^−^ Tregs proportion in HE4-positive (>140 pmol/L) OC patients was also higher than that in HE4-negative patients (Table 6; *P <* 0.05). In addition, we examined the correlation between CD4^+^ Treg, CD8^+^Treg, CD3^+^ T cell subsets, and CA125, HE4. Interestingly, CD4^+^Foxp3^+^, CD8^+^CD28^−^, CD8^+^CD122^+^ Treg subsets, and CD3^+^Vδ1 T cells were all positively correlated with CA125 (r = 0.6221, *P <* 0.0001; r = 0.4999, *P* = 0.0019; r = 0.4705, *P* = 0.0038; r = 0.5212, *P* = 0.0011), and CD4^+^Foxp3^+^ Tregs and CD3^+^Vδ1 T cell subsets were both positively correlated with HE4 (r = 0.4403, *P* = 0.0072; r = 0.4165, *P* = 0.0115) (Figure 3b–d).

### 3.4. The Relationship between Proportion of Different Treg Subsets and TGF-β of OC Patients

In addition to the proportion changes of the Treg subsets, OC patients showed vigorous TGF-β levels in serum. The results of the ELISA indicated that TGF-β increased significantly in OC patients (664.44 ± 232.72) when compared with BOT patients (389.09 ± 143.46) and HC (265.45 ± 160.45) (Figure 4a, *P <* 0.05). Previous studies have shown that high levels of TGF-β may be involved in the induction of Treg production [18,19]. We further analyzed the relationship among CD4^+^Treg, CD8^+^Treg, CD3^+^Vδ1 T, CD3^+^Vδ2 T cell subsets, and TGF-β. As shown in Figure 4c–e,g,h, we found that CD4^+^Foxp3^+^ (r = 0.3562, *P <* 0.0001), CD4^+^CD25^+^Foxp3^+^ (r = 0.5499, *P* = 0.0330), CD8^+^CD28^−^ (r = 0.5499, *P* = 0.0005), CD8^+^CD28^−^Foxp3^+^ (r=0.3637, *P*=0.0292), and CD8^+^CD122^+^ (r = 0.5502, *P* = 0.0005) Treg subsets were all positively correlated with TGF-β. No significant association was found among CD4^+^CD25^+^ Tregs, CD8^+^Foxp3^+^ Tregs, CD3^+^Vδ1 T cells, CD3^+^Vδ2 T cell subsets, and TGF-β (Figure 4b,f,i,j; *P >* 0.05).

## 4. Discussion

Tregs are frequently infiltrated in the tumor microenvironment, about 10 to 50%, as opposed to 2 to 5% in nontumor individuals [20]. In previous studies, we and others have shown elevated levels of Tregs in tumors tissues and peripheral blood of patients with OC [21,22,23]. Notably, elevated expression of some molecular markers on the surface of T cells can be used to distinguish Tregs, including interleukin-2 receptor α chain (CD25), CD127, and CD28. Human CD4^+^ Tregs express high levels of the CD25 and the forkhead winged-helix transcription factor (Foxp3), which are pivotal for their development and function [24]. However, in most studies, only one specific Treg subpopulation was analyzed. Our data first show a higher level of CD4^+^Treg subsets (CD4^+^Foxp3^+^ and CD4^+^CD25^+^Foxp3^+^) in OC groups than in BOT and HC groups (Figure 1), suggesting an immunosuppression potential of these subsets in patients with OC. Additionally, CD8^+^ Tregs were the first identified cell subset with a suppressive potential in 1972 [25], and they are associated with different phenotypes depending on the studies (CD28, CD122) [26,27,28,29]. Our studies have examined how CD8^+^Treg subpopulations in tumor tissues and peripheral blood contribute to the prognosis of OC [12,29]. However, the key Treg subsets in patients with OC remain disputed. Here, we found the difference among CD8^+^ Treg subpopulations (CD8^+^CD28^−^, CD8^+^Foxp3^+^, CD8^+^Foxp3^+^CD28^−^, CD8^+^CD122^+^) in OC patients compared with BOT and HC groups (Figure 1). Interestingly, the downregulated expression of CD28, not the upregulated expression of Foxp3 in CD8^+^ Tregs, have stronger resolution potential between OC patient and BOT patients or HC. Therefore, we speculate that Tregs share many features, but possess distinct differences according to cancer type.

More and more studies have suggested that γδ T cells also have immunosuppressive effects on TME [30,31]. Our previous study has also shown a higher level of tumor-infiltrating γδ T cells and CD3^+^γδ^+^Vδ1 T cells in OC patients than in paired BOT patients and normal ovarian tissue, suggesting a poor prognosis in OC patients [32,33,34]. Our results are consistent with these studies, but the primary elevated γδ T cell subsets were CD3^+^Vδ1 T cell populations in peripheral blood (Figure 1). These results suggested that CD3^+^Vδ1 T cell subsets have a strong ability to distinguish OC patients from BOT patients and HC. Overall, analysis of the Treg subpopulations in peripheral blood is undeniably important in studies of different cancer immunology.

Combined with the above results, we proved that the high abundance of CD4^+^ Treg and CD8^+^ Treg subpopulations but not CD3^+^γδ T cell subpopulations were associated with FIGO stage, lymph node metastasis, and distant metastasis status at the diagnosis of OC patients (Table 3, Table 4 and Table 5). Our study is the first to show a positive association between αβ and γδT cells and OC stage/metastatic status, although many studies have revealed a correlation between Tregs in tumor lesions and patient prognosis. Okla K. et al. demonstrated the increased frequency of M-MDSC in the tumor lesions in EOC and its correlation with stage, but not Tregs [35]. Other studies showed that a higher level of CD30^+^OX40^+^ Tregs were associated with improved overall survival, whereas CD39^+^γδ Tregs were associated with poor prognosis in colorectal cancer. These results confirm the tremendous heterogeneity of the clinical relevance of Treg subpopulations in human cancers.

CA125 (also known as mucin 16) and human epididymis protein 4 (HE4; also known as WFDC2) are often used to screen benign and malignant pelvic tumors. Both CA125 and HE4 are tumor markers associated with the ovary. In this study, we observed elevated levels of CA125 and HE4 in the serum of patients with OC compared to BOT patients and HC (Figure 3a). It is worth noting that the diagnosis of ovarian cancer usually occurs in the late stage of the disease. Importantly, >80% of patients have asymptomatic tumor recurrence, and recurrent OC is most often detected by elevated levels of CA125 [29]. However, not all patients with recurrent tumors have elevated serum CA125 levels, and early detection of recurrence by detecting CA125 levels cannot evaluate the prognosis of patients. In these patients, alternative biomarkers, such as HE4, might be of use for the monitoring of recurrent cancer, but this needs further evaluation. Here, we assessed the potential clinical relevance of Treg and γδ T cell subpopulations in monitoring the recurrence of OC and demonstrated the positive correlations between these cell subpopulations and CA125 or HE4 (Figure 3b–d, Table 6). The use of Treg and γδ T cell subpopulations might be potential biomarkers for OC monitoring, but we need to do further exploration to confirm the above hypothesis.

Tregs exert their immunosuppressive activity by secreting various cytokines, and a high level of TGF-β has been identified as a marker of advanced malignancy and poor overall prognosis in a variety of malignancies, including OC [36,37,38]. TGFβ released by cancer cells in TME promotes cancer progression by shaping the architecture of the tumor and by suppressing the antitumor activities of immune cells, thus generating an immunosuppressive environment that prevents or attenuates the efficacy of anticancer immunotherapies. The repression of TGFβ signaling is therefore considered a prerequisite and major avenue to enhance the efficacy of current and forthcoming immunotherapies [39]. TGF-β promotes T cell differentiation into Tregs [40] and enables Tregs to inhibit adaptive and innate immune responses. CD4^+^ T cells can acquire cytotoxic activity when TGF-β signaling is inactivated [41]. Importantly, CD4^+^ T cells exhibit considerable plasticity in TME, depending on the cytokine environment, specifically TGF-β levels [42]. Consistent with our previous reports [43], we confirmed that the levels of serum TGF-β in OC patients was higher than in BOT patients and HC here. Tregs exert an immunosuppressive function mainly by secreting cytokines to inhibit T cell proliferation and downregulate the immune function of Th1 cells, but the link between Tregs and tumor cytokine signaling remains largely unexplored. We found that the TGF-β-induced p38 MAPK signaling pathway contributes to the activation of CD8^+^ Tregs in the OC microenvironment, suggesting Tregs respond to TGF-β, and high levels of TGF-β were positively associated with the proportion of CD4^+^ Tregs and CD8^+^ Tregs but not γδ T cell subpopulations (Figure 4b–j), suggesting Treg subsets could be a robust indicator of OC patient survival. Whereas monitoring disease progression in patients with OC by TGF-β levels, TGF-β inhibition is combined with immune checkpoint inhibition to affect the production of Tregs, thereby targeting the immunosuppressive microenvironment, ultimately breaking immune tolerance and improving immunotherapy efficacy.

## 5. Conclusions

In summary, we showed that circulating CD4^+^Treg, CD8^+^Treg, and CD3^+^γδ T cell subpopulations presented in OC patients at a significantly higher level than BOT patients and HC, and there is a positive correlation with FIGO stage/metastatic status, CA125, HE4, and TGF-β. Specifically, Treg phenotype molecules, including Foxp3 in CD4^+^ Tregs, CD28 in CD8^+^ Tregs, and Vδ1 in CD3^+^γδ T cells exert a significant difference. These results provide a piece of preliminary evidence that the proportion of circulating Treg subsets has represented a promising marker for the immunologic surveillance of OC.

## Figures and Tables

**Figure 1 medicina-59-00205-f001:**
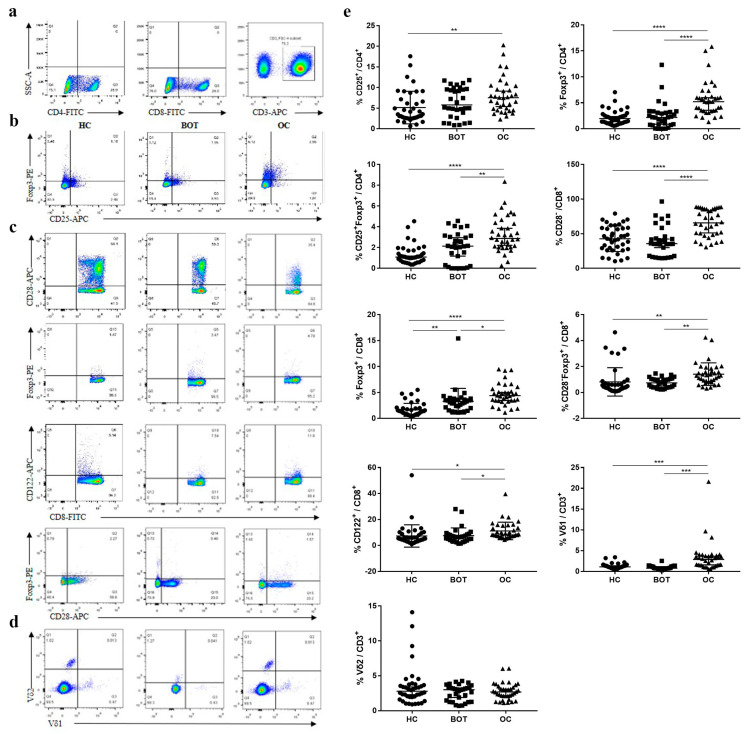
Increased proportion of CD4^+^ Tregs, CD8^+^ Tregs, and CD3^+^γδ T cell subsets in peripheral blood of OC patients. (**a**) Gating strategy for the analysis of Tregs by FCM; (**b**) representative dot plots CD25 and Foxp3 expression on CD4^+^ Tregs from HC (*n* = 40), BOT patients (*n* = 32), and OC patients (*n* = 36) are shown; (**c**) representative dot plots CD28, Foxp3, and CD122 expression on CD8^+^ Tregs from HC, BOT patients, and OC patients are shown; (**d**) representative dot plots Vδ1 and Vδ2 expression on CD3^+^γδT cells from HC, BOT patients, and OC patients are shown; (**e**) proportions of circulating CD4^+^ Tregs, CD8^+^ Tregs, and CD3^+^γδ T cell subsets in HC, BOT patients, and OC patients.* *P <* 0.05, ** *P <* 0.01, *** *P <* 0.001, **** *P <* 0.0001.

**Figure 2 medicina-59-00205-f002:**
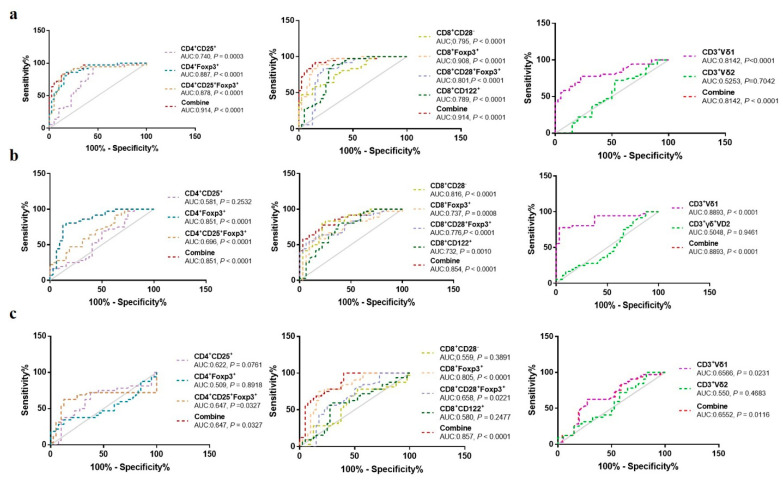
ROC curves of circulating CD4^+^Treg, CD8^+^Treg, and CD3^+^γδ T cell subsets in distinguishing OC patients, BOT patients, and HC. (**a**) OC vs. HC; (**b**) OC vs. BOT; (**c**) BOT vs. HC.

**Figure 3 medicina-59-00205-f003:**
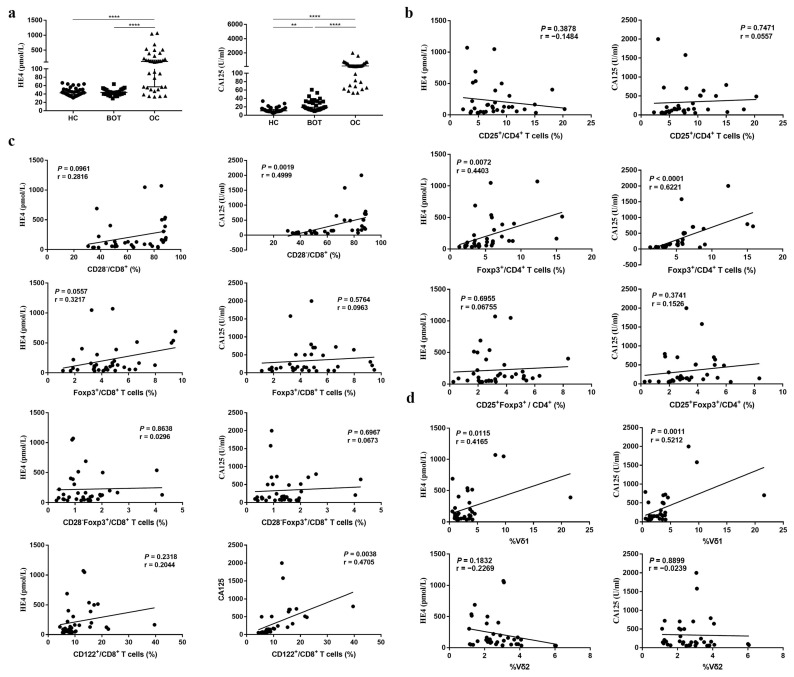
The association between CD4^+^ Tregs, CD8^+^ Tregs, and CD3^+^γδ T cells proportion and HE4, CA125 in OC patients. (**a**) HE4, CA125 levels in serum of OC patients, BOT patients, and HC. (**b**–**d**). The association between CD4^+^ Tregs, CD8^+^ Tregs, CD3^+^γδ T cells proportion and HE4, CA125. ** *P <* 0.01, **** *P <* 0.0001.

**Figure 4 medicina-59-00205-f004:**
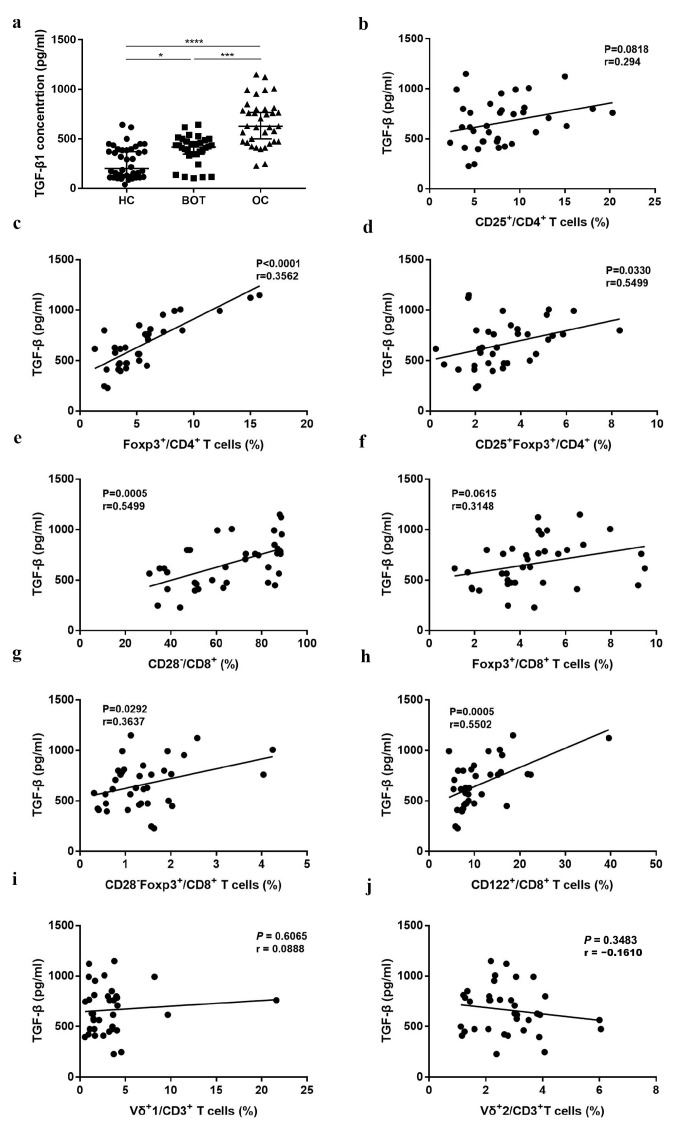
The association among the CD4^+^ Tregs, CD8^+^ Tregs, CD3^+^γδ T cells proportion, and TGF-β in OC patients. (**a**) TGF-β levels in serum of OC patients, BOT patients, and HC. (**b**–**j**) The association among CD4^+^CD25^+^, CD4^+^Foxp3^+^, CD4^+^CD25^+^Foxp3^+^, CD8^+^CD28^−^, CD8^+^Foxp3^+^, CD8^+^CD28^−^Foxp3^+^, CD8^+^CD122^+^ Tregs and CD3^+^Vδ1 T cells, and CD3^+^Vδ2 T cells and TGF-β. * *P* < 0.05, *** *P* < 0.001, **** *P* < 0.0001.

**Table 1 medicina-59-00205-t001:** Characteristic features of ovarian cancer patients (*n* = 36).

Variables	Patients, *n*	%
Age (year)		
<50	14	38.9
≥50	22	61.1
Tumor size (cm)		
<5	4	11.1
≥5	32	88.9
Histology		
serous	22	61.1
mucinous	1	2.8
endometrioid	4	11.1
clear cell carcinoma	8	22.2
others	1	2.8
FIGO stage		
I–II	18	50.0
III–IV	18	50.0
Lymph node metastasis		
No	21	58.3
Yes	15	41.7
Distant metastasis		
No	19	52.8
Yes	17	47.2
CA125 (U/mL)		
≤35	0	0.0
>35	36	100.0
HE4 (pmol/L)		
≤140	22	61.1
>140	14	38.9

CA125: Carbohydrate antigen 125; HE4: human epididymis protein 4; FIGO: federation of gynecology obstetrics.

**Table 2 medicina-59-00205-t002:** Treg subsets levels (%) in ovarian cancer patients (OC), benign ovarian tumor patients (BOT), and healthy controls (HC).

Variables	OC	BOT	HC	*P*-Value
CD4^+^ Tregs				
CD4^+^CD25^+, ††^	8.03 ± 4.24	6.39 ± 3.63	5.15 ± 3.95	0.008
CD4^+^Foxp3^+,^ ****^, ††††^	5.62 ± 3.31	2.42 ± 2.53	2.15 ± 1.37	<0.0001
CD4^+^CD25^+^Foxp3^+,^ **^, ††††^	3.28 ± 1.70	2.00 ± 1.46	1.32 ± 0.91	<0.0001
CD8^+^ Tregs				
CD8^+^CD28^−,^ ****^, ††††^	65.61 ± 19.75	39.39 ± 21.25	42.73 ± 1.84	<0.0001
CD8^+^Foxp3^+,^ *^, †††^	4.63 ± 2.09	3.31 ± 2.47	1.66 ± 1.25	<0.0001
CD8^+^CD28^−^Foxp3^+,^ **^, ††^	1.41 ± 0.87	0.74 ± 0.37	0.82 ± 1.09	0.002
CD8^+^CD122^+,^ *^, †^	11.18 ± 6.73	7.56 ± 5.91	7.28 ± 8.59	0.043
CD3^+^ γδ T cells				
CD3^+^Vδ1 T, ***^, †††^	3.36 ± 3.68	0.92 ± 0.44	1.22 ± 0.65	<0.0001
CD3^+^Vδ2 T	2.76 ± 1.20	2.59 ± 2.57	3.43 ± 4.82	0.058

**OC:** ovarian cancer patients, **BOT:** benign ovarian tumor patients, **HC:** healthy controls. * *P* <0.05, ** *P <* 0.01, *** *P <* 0.001, **** *P <* 0.0001, OC vs. BOT. ^†^
*P <* 0.05, ^††^
*P <* 0.01, ^†††^
*P <* 0.001, ^††††^
*P <* 0.0001, OC vs. HC. One-way ANOVA was used.

**Table 3 medicina-59-00205-t003:** The association between the CD4^+^ Tregs proportion (%) and clinical characteristics of ovarian cancer patients.

Variables	CD4^+^CD25^+^	*P*-Value	CD4^+^Foxp3^+^	*P*-Value	CD4^+^CD25^+^Foxp3^+^	*P*-Value
Age (year)						
<50	7.33 ± 5.15	0.44	4.94 (2.90–9.70)	0.365	2.43 ± 1.57	0.015 *
≥50	8.47 ± 3.61		5.20 (3.54–6.08)		3.81 ± 1.58	
Tumor size (cm)						
<5	7.01 ± 1.40	0.618	4.74 ± 1.76	0.58	2.85 (2.62–3.14)	0.195
≥5	8.15 ± 4.47		8.15 ± 4.47		3.00 (2.03–4.60)	
Histology						
serous	8.64 ± 4.70	0.528	5.96 ± 3.26	0.205	3.55 ± 1.74	0.267
endometrioid	8.17 ± 5.22		7.33 ± 5.20		2.75 ± 1.90	
clear cell carcinoma	7.20 ± 2.40		4.79 ± 1.96		3.30 ± 1.35	
others	4.30 ± 0.97		1.72 ± 0.58		1.19 ± 1.31	
FIGO stage						
I–II	6.52 ± 3.08	0.031 *	4.50 ± 3.01	0.040 *	2.40 (2.01–3.01)	0.003 *
III–IV	9.53 ± 4.77		6.74 ± 3.29		3.86 (2.68–5.27)	
Lymph node metastasis						
No	7.34 ± 3.68	0.256	4.48 ± 2.81	0.012	2.57 (2.03–3.23)	0.011 *
Yes	8.99 ± 4.89		7.21 ± 3.37		3.87 (2.81–5.40)	
Distant metastasis						
No	5.38 (4.33–7.57)	0.001 *	3.99 (3.09–5.20)	0.002 *	2.57 (2.03–2.94)	0.003 *
Yes	9.55 (7.58–14.10)		6.04 (5.47–8.93)		3.87 (2.74–5.31)	

FIGO: Federation of Gynecology Obstetrics. * *P* < 0.05 indicates a statistically significant difference. Independent-sample *t*-test, one-way ANOVA, and Mann–Whitney U were used.

**Table 4 medicina-59-00205-t004:** The association between the CD8^+^ Treg subsets proportion (%) and clinical characteristics of ovarian cancer patients.

Variables	CD8^+^CD28^−^	*P*-Value	CD8^+^Foxp3^+^	*P*-Value	CD8^+^CD28^−^Foxp3^+^	*P*-Value	CD8^+^CD122^+^	*P*-Value
Age (year)								
<50	63.73 ± 21.87	0.655	4.80 (1.88–6.96)	0.611	1.21 (0.64–2.17)	0.417	10.55 (6.98–17.45)	0.162
≥50	66.81 ± 18.72		4.30 (3.43–5.03)		1.32 (0.88–1.68)		8.66 (7.20–10.62)	
Tumor size (cm)								
<5	66.50 ± 15.72	0.926	3.4 ± 1.60	0.216	0.84 ± 0.48	0.171	9.65 ± 4.11	0.635
≥5	65.50 ± 20.41		4.78 ± 2.11		1.48 ± 0.89		11.37 ± 7.01	
Histology								
serous	70.43 ± 19.26	0.059	4.17 ± 1.73	0.259	4.71 ± 1.73	0.878	10.92 ± 5.23	0.481
endometrioid	67.72 ± 16.56		3.74 ± 0.71		3.74 ± 0.71		17.39 ± 15.02	
clear cell carcinoma	59.05 ± 17.89		5.43 ± 3.08		5.43 ± 3.08		10.19 ± 3.94	
others	34.60 ± 0.56		2.30 ± 1.66		2.30 ± 1.66		5.61 ± 0.32	
FIGO stage								
I–II	57.77 ± 20.72	0.015 *	4.34 ± 2.19	0.41	1.30 ± 0.30	0.453	10.54 ± 7.90	0.573
III–IV	73.46 ± 15.60		4.92 ± 2.00		1.52 ± 1.08		11.83 ± 5.46	
Lymph node metastasis								
No	59.93 ± 20.11	0.039 *	4.21 ± 2.09	0.157	1.23 ± 0.60	0.139	10.08 ± 7.40	0.249
Yes	73.56 ± 16.78		5.22 ± 2.01		1.67 ± 1.21		12.73 ± 5.53	
Distant metastasis								
No	57.40 ± 20.59	0.006 *	4.36 ± 2.56	0.416	1.32 ± 0.86	0.518	8.01 (7.03–9.96)	0.056
Yes	74.78 ± 14.39		4.93 ± 1.42		1.51 ± 0.90		10.30 (7.86–17.30)	

FIGO: Federation of Gynecology Obstetrics. * *P* < 0.05 indicates a statistically significant difference. Independent-sample *t*-test, one-way ANOVA, and Mann–Whitney U were used.

**Table 5 medicina-59-00205-t005:** The association between CD3^+^γδ T cell subsets proportion (%) and clinical characteristics of ovarian cancer patients.

Variables	CD3^+^Vδ1	*P*-Value	CD3^+^Vδ2	*P*-Value
Age (year)				
<50	2.92 ± 2.03	0.58	3.16 ± 1.48	0.105
≥50	3.63 ± 4.45		2.50 ± 0.92	
Tumor size (cm)				
<5	6.99 ± 9.75	0.464	2.29 ± 0.95	0.42
≥5	2.90 ± 2.04		2.82 ± 1.23	
Histology				
serous	3.84 ± 4.28	0.782	2.42 ± 0.91	0.404
endometrioid	3.17 ± 4.35		3.20 ± 0.49	
clear cell carcinoma	2.26 ± 1.38		3.01 ± 1.59	
others	2.84 ± 1.80		4.53 ± 2.14	
FIGO stage				
I–II	3.16 ± 4.75	0.752	2.82 ± 1.43	0.789
III–IV	3.56 ± 2.89		2.70 ± 0.96	
Lymph node metastasis				
No	3.16 ± 4.38	0.707	2.83 ± 1.36	0.676
Yes	3.64 ± 2.51		2.66 ± 0.98	
Distant metastasis				
No	3.24 ± 4.59	0.845	2.66 ± 1.41	0.6
Yes	3.49 ± 2.43		2.87 ± 0.95	

FIGO: Federation of Gynecology Obstetrics. Independent sample *t*-test, one-way ANOVA were used.

**Table 6 medicina-59-00205-t006:** The association between CD4^+^ Treg, CD8^+^ Treg, and CD3^+^γδ T cell subsets proportion (%) and tumor markers.

Variables	CA125 (U/mL)		*P*-Value	HE4 (pmol/L)		*P*-Value
	<Median	>Median		≤140	>140	
CD4^+^CD25^+^	7.13 ± 4.10	9.02 ± 4.29	0.186	8.15 ± 4.28	7.84 ± 4.33	0.835
CD4^+^Foxp3^+^	3.87 ± 1.95	7.57 ± 3.46	0.000 *	4.02 (2.90–5.90)	6.05 (5.21–9.84)	0.008
CD4^+^CD25^+^Foxp3^+^	2.86 ± 1.88	3.74 ± 1.37	0.122	3.18 ± 1.65	3.42 ± 1.81	0.683
CD8^+^CD28^−^	50.50 (38.40–60.40)	85.80 (77.85–88.05)	0.000 *	58.78 ± 16.85	76.35 ± 19.71	0.007 *
CD8^+^Foxp3^+^	3.90 ± 2.01	5.44 ± 1.92	0.025 *	4.19 (3.44–5.05)	4.88 (3.24–7.38)	0.114
CD8^+^CD28^−^Foxp3^+^	1.12 ± 0.56	1.74 ± 1.05	0.032 *	1.36 ± 0.83	1.49 ± 0.95	0.653
CD8^+^CD122^+^	7.51 (6.31–8.05)	15.20 (10.13–17.80)	0.000 *	9.14 ± 4.75	14.39 ± 8.20	0.020
CD3^+^Vδ1	1.57 (1.01–2.56)	3.71 (3.39–4.31)	0.013 *	1.88 (1.40–3.58)	3.64 (1.48–5.11)	0.103
CD3^+^Vδ2	2.30 ± 1.36	2.49 ± 0.96	0.208	3.01 ± 1.33	2.36 ± 0.87	0.114

* *P* < 0.05 indicates a statistically significant difference. Independent sample *t*-test and Mann–Whitney U were used.

## Data Availability

Not applicable.

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
