# Peer review of "Circulating CD4+ Treg, CD8+ Treg, and CD3+ γδ T Cell Subpopulations in Ovarian Cancer"

_medicina, 2023, doi:10.3390/medicina59020205_

Round 1
Reviewer 1 Report
The manuscript submitted for review concerns the evaluation of regulatory T cells in patients with ovarian cancer and women in the control groups.
My comments on the manuscript.
1. It was not explained at first appearance what the abbreviations mean, e.g.: OC, BOT (whether borderline or benign) HC, TME, FMC etc.
2. The spelling of the CA12-5 marker is also incorrect
3. Citing literature in the first part of the work completely illegible, later in superscript, later in square brackets.
4. Many words in the text are separated by a hyphen - probably an editorial error. E.g.: ex-pected, ma-lignancies.
5. There is no group division explained. For example, why patients <50 years of age or tumors smaller than 5 cm.
6. Tables and especially figures are very complex and therefore illegible.
7. I wonder what this research adds to the knowledge about ovarian cancer. I guess not much.
8. The form of transferring knowledge about possibly interesting research is difficult to accept
I believe that the manuscript is not suitable for publication in this form.
Author Response
Dear Ms.Winifred Zhao and Reviewers:
Thank you very much for your kindly reply on our manuscript entitled “Circulating CD4+ Treg, CD8+ Treg and CD3+ γδT cells subpopulations in ovarian cancer” (medicina-2057434). We appreciate your time and suggestions. All the comments and suggestions raised in your letter are valuable and of important significance on guiding us to be more considerate in present and future work. We have done some modifications on the writing in our manuscript and mentioned all of them in this letter. The suggestions are of great help for us to improve our manuscript. We made revisions by using a yellow font color or a yellow background to flag revised positions which were changed in the manuscript. We sincerely hope that our revised manuscript could be approved. In addition, a point-by-point reply to the reviewers’ concerns is included below.
Reviewers’ Comments:
Referee #1 (Remarks to the Author):
Major comments:
1. It was not explained at first appearance what the abbreviations mean, e.g.: OC, BOT (whether borderline or benign) HC, TME, FMC etc.
Response: Thank you for these comments. They were very critical and helpful for improving our work. We supplement all abbreviations for the first time, and the revised content is shown in the yellow part of the manuscript (lines17-31, page 1).
2. The spelling of the CA125 marker is also incorrect
Response: Thank you for these comments. They were very critical and helpful for improving our work. We have corrected the spelling of the CA125 in manuscript, and the revised content is shown in the yellow part of the manuscript.
3. Citing literature in the first part of the work completely illegible, later in superscript, later in square brackets.
Response: Thank you for these comments. They were very critical and helpful for improving our work. We added references to this section and go through the manuscript carefully and provide citations for all statements provided. The revised content is shown in the references part of the manuscript (lines 45-52, page 2).
4. Many words in the text are separated by a hyphen - probably an editorial error. E.g.: ex-pected, ma-lignancies.
Response: Thank you for these comments. They were very critical and helpful for improving our work. We have deleted all unnecessary hyphenations in the manuscript and checked other formatting and syntax errors. However, we are very sorry that due to the format requirements of the manuscript, indent characters are required for each line of the manuscript, and many words cannot be fully displayed in the same line. We have tried to adjust many times and optimized some words, and the revised content is shown in the yellow part of the manuscript.
5. There is no group division explained. For example, why patients <50 years of age or tumors smaller than 5 cm.
Response: Thank you for these comments. They were very critical and helpful for improving our work. The grouping in the manuscript table is based on the references we consulted, and the references we cited have been added to the manuscript. The revised content is shown in the yellow part of the manuscript (line 95, page 3).
6. Tables and especially figures are very complex and therefore illegible.
Response: Thank you for these comments. They were very critical and helpful for improving our work. We have revised the format and content of all the tables, thanks for the experts' correction. The revised content is shown in the yellow part of the manuscript (page 7, 9-11, 14).
7. I wonder what this research adds to the knowledge about ovarian cancer. I guess not much.
Response: Thank you for these comments. They were very critical and helpful for improving our work. At present, there is no obvious curative effect in the treatment of OC, such as surgery, chemotherapy and immunosuppressive drugs. To understand the tumor immunosuppressive microenvironment may provide a promising path for the treatment of OC. In addition, immune monitoring during tumor treatment (no matter surgery or chemotherapy, etc.) can provide relevant evidence for treatment effect and patient prognosis. In previous studies, we explored the significance of CD4+Treg and CD8+Treg in ovarian cancer, and we also found that γδT cells also have immunosuppressive subtypes associated with OC. Therefore, we believe that the targeted evaluation of OC immunosuppressive cells, especially Treg cells, should not only be CD4+ Tregs (most studies may mainly focus on CD4+CD25+ Tregs), and should conduct comprehensive evaluation of each subgroup of Tregs, to monitor the immunosuppression of OC. It may also provide some clinical basis for future treatment based on Tregs. In addition, there are many types of Tregs. In this paper, we will report that the main types of CD4Tregs, CD8Tregs and γδ tregs have been studied in OC, to clarify which types of Tregs are associated with the clinical characteristics of OC. Of course, there are still some defects in this study, and we will continue to improve future studies according to the opinions of experts.
8. The form of transferring knowledge about possibly interesting research is difficult to accept.
Response: Thank you for these comments. They were very critical and helpful for improving our work. It was a weakness of our manuscript that we didn’t transfer knowledge in a simple way. We will consider this point in follow-up research. Therefore, thank you again for your valuable suggestions, so that we have an opportunity to make amends in future.
We would like to thank the reviewers and editors again for your positive and helpful comments and for the opportunity to revise our manuscript by incorporating these recommended changes.
We are looking forward to your positive decision!
Sincerely,
Fang Wang, MD., PhD.
Department of Laboratory Medicine
the First Affiliated Hospital of Nanjing Medical University
Nanjing, China
Phone: +86-86862814
Email: [email protected]
Reviewer 2 Report
The authors have shown that circulating T-cell populations show strong correlations with OC stage and is predictive of its diagnosis. Recent studies have shown that OC is closely associated with a marked upregulation of immunesuppression and the enrichment of certain immune suppressive cell types in the immune mileu. Identifying these profiles are essential for strategizing the type of treatment that patients could undergo. This paper provides important information regarding this. The paper is well written, but however many formatting and syntax errors is prevalent throughout the manuscript. Please correct them prior to publishing. It would also be informative if authors could provide a paragraph on the effects of various chemotherapeutic interventions (published data) on Treg populations in OC patients and how their findings could help advance immunotherapy and other therapeutic strategies in OC treatment.
1. remove unnecessary hyphenations throughout manuscriopt. For e.g. “ma-lignancies” on page 1 should be “malignancies”. There are several such incidences throughout the manuscript.
2. Provide citations for statements in section 3.3. Paragraph 1. Please go through the manuscript carefully and provide citations for all statements provided. It is often not provided in the manuscript.
3. Tables have to be reformatted to fit columns and rows. Very poorly done.
4. The association between TGFb and Treg cells are quite interesting. Please discuss this in relation to immunotherapy prospects in discussion
Author Response
Dear Ms.Winifred Zhao and Reviewers:
Thank you very much for your kindly reply on our manuscript entitled “Circulating CD4+ Treg, CD8+ Treg and CD3+ γδT cells subpopulations in ovarian cancer” (medicina-2057434). We appreciate your time and suggestions. All the comments and suggestions raised in your letter are valuable and of important significance on guiding us to be more considerate in present and future work. We have done some modifications on the writing in our manuscript and mentioned all of them in this letter. The suggestions are of great help for us to improve our manuscript. We made revisions by using a yellow font color or a yellow background to flag revised positions which were changed in the manuscript. We sincerely hope that our revised manuscript could be approved. In addition, a point-by-point reply to the reviewers’ concerns is included below.
Reviewers’ Comments:
Referee #2 (Remarks to the Author):
1. Major comments:
The authors have shown that circulating T-cell populations show strong correlations with OC stage and is predictive of its diagnosis. Recent studies have shown that OC is closely associated with a marked upregulation of immunesuppression and the enrichment of certain immune suppressive cell types in the immune mileu. Identifying these profiles are essential for strategizing the type of treatment that patients could undergo. This paper provides important information regarding this. The paper is well written, but however many formatting and syntax errors is prevalent throughout the manuscript. Please correct them prior to publishing. It would also be informative if authors could provide a paragraph on the effects of various chemotherapeutic interventions (published data) on Treg populations in OC patients and how their findings could help advance immunotherapy and other therapeutic strategies in OC treatment.
Response: Thank you for your kind and helpful comments and for the opportunity to revise our manuscript. Special thanks to you! We have added relevant content in introduction and colored this section yellow according to your suggestion (lines 61-71, page 2).
2. Main Suggestions:
- Remove unnecessary hyphenations throughout manuscriopt. For e.g. “ma-lignancies” on page 1 should be “malignancies”. There are several such incidences throughout the manuscript.
Response: Thank you for these comments. They were very critical and helpful for improving our work. We have deleted all unnecessary hyphenations in the manuscript and checked other formatting and syntax errors. However, we are very sorry that due to the format requirements of the manuscript, indent characters are required for each line of the manuscript, and many words cannot be fully displayed in the same line. We have tried to adjust many times and optimized some words. The revised content is shown in the yellow part of the manuscript.
- Provide citations for statements in section 3.3. Paragraph 1. Please go through the manuscript carefully and provide citations for all statements provided. It is often not provided in the manuscript.
Response: Thank you for these comments. They were very critical and helpful for improving our work. We have provided citations for statements in section 3.3. Paragraph 1, and the references in the manuscript were supplemented and corrected. The revised content is shown in the references part of the manuscript (line 224-225, page 12).
- Tables have to be reformatted to fit columns and rows. Very poorly done.
Response: Thank you for these comments. They were very critical and helpful for improving our work. We have revised the format and content of all the tables in the manuscript, thanks for the experts' correction. The revised content is shown in the yellow part of the manuscript (page 7, 9-11, 14).
- The association between TGFb and Treg cells are quite interesting. Please discuss this in relation to immunotherapy prospects in discussion.
Response: Thank you for these comments. They were very critical and helpful for improving our work. We have added the relationship between TGFb and Treg cells in the discussion section and further explored it in combination with immunotherapy for ovarian cancer. The supplementary content is shown in the yellow part of the manuscript (lines368-382, 394-402, page 18-19).
We would like to thank the reviewers and editors again for your positive and helpful comments and for the opportunity to revise our manuscript by incorporating these recommended changes.
We are looking forward to your positive decision!
Sincerely,
Fang Wang, MD., PhD.
Department of Laboratory Medicine
the First Affiliated Hospital of Nanjing Medical University
Nanjing, China
Phone: +86-86862814
Email: [email protected]
Reviewer 3 Report
Abstract:
Conclusion:”These indicators have the potential to be used as immunosurveillance biomarkers for OC” What were the sensitivity, specificity, and accuracy of indicators? Is it possible to be reported?
Introduction:
Please discuss about the burden of OC in the world and use the most update articles: PMID: 36439044
FCM? Before using the abbreviation please introduce the full-text of the words.
Last paragraph introduction:
“We first confirmed CD4+ Treg cells, CD8+ Treg cells, CD3+Vδ1 T cells and CD3+Vδ2 T cells subpopula-tions in the blood of the OC patients using FCM. We then plotted receiver operating characteristic (ROC) curves to explore the abil-ity of Treg cells subpopulations to distinguish between OC pa-tients, BOT patients, and healthy controls. Moreover, we studied the relationship between different Treg cells subpopulations and clinical characteristics, tumor biomarkers (CA12-5, HE4) and TGF-β.”
Should be moved to methods and materials section at the appropriate place.
Methods and materials:
Needs to be revised critically. Please note to comments below:
What is the type of study?
What was the inclusion criteria of study?
When was blood sampling done? Where was blood sampling done?
EDTA??
Was flow cytometry done for first group (OC group) not for BOT and control groups?
Statistical analysis:
Please revives statistical analysis section. “Qualitative variables were reported as percentages and quantitative variables were reported with mean and standard deviation”.
The differences between groups were compared using an independent-samples t test?, one-way ANOVA ?and Mann-Whitney U tests test. For which groups mentioned test were used?
Results:
The proportion of CD4+CD25+ Treg cells was higher in OC patients than in healthy controls (8.02 ± 4.24 vs 5.15 ± 3.95, P<0.05). How about the BOT groups?
Result section is difficult to be followed. Please draw a table and post some of information and data of three groups (like proportion of Treg cells (Foxp3+ , CD25+Foxp3+ ) in CD4+ T cells, …)
Author Response
Dear Ms.Winifred Zhao and Reviewers:
Thank you very much for your kindly reply on our manuscript entitled “Circulating CD4+ Treg, CD8+ Treg and CD3+ γδT cells subpopulations in ovarian cancer” (medicina-2057434). We appreciate your time and suggestions. All the comments and suggestions raised in your letter are valuable and of important significance on guiding us to be more considerate in present and future work. We have done some modifications on the writing in our manuscript and mentioned all of them in this letter. The suggestions are of great help for us to improve our manuscript. We made revisions by using a yellow font color or a yellow background to flag revised positions which were changed in the manuscript. We sincerely hope that our revised manuscript could be approved. In addition, a point-by-point reply to the reviewers’ concerns is included below.
Reviewers’ Comments:
Referee #3 (Remarks to the Author):
1. (Abstract): Conclusion: “These indicators have the potential to be used as immunosurveillance biomarkers for OC” What were the sensitivity, specificity, and accuracy of indicators? Is it possible to be reported?
Response: Thank you for these comments. They were very critical and helpful for improving our work. The sensitivity and specificity of each Tregs subpopulation in identifying ovarian cancer have been shown in Figure 2 and summarized in the conclusion. In ROC analysis, the AUC value range is 0-1. The closer the AUC is to 1, the stronger the discrimination ability of this indicator is.
2. (Introduction): Please discuss about the burden of OC in the world and use the most update articles: PMID: 36439044.
Response: Thank you for these comments. They were very critical and helpful for improving our work. We have read these papers (include PMID: 36439044) and discussed them in the introduction in lines 45-52, page 2. We also cited the refs in Reference 1-3.
3. (Introduction): FCM? Before using the abbreviation, please introduce the full text of the words.
Response: Thank you for these comments. They were very critical and helpful for improving our work. flow cytometry (FCM). We supplement all abbreviations for the first time, and the revised content is shown in the yellow part of the manuscript (lines 17-31, page 1).
4. (Introduction): Last paragraph introduction:“We first confirmed CD4+ Treg cells, CD8+ Treg cells, CD3+Vδ1 T cells and CD3+Vδ2 T cells subpopula-tions in the blood of the OC patients using FCM. We then plotted receiver operating characteristic (ROC) curves to explore the abil-ity of Treg cells subpopulations to distinguish between OC pa-tients, BOT patients, and healthy controls. Moreover, we studied the relationship between different Treg cells subpopulations and clinical characteristics, tumor biomarkers (CA125, HE4) and TGF-β.” Should be moved to methods and materials section at the appropriate place.
Response: Thank you for these comments. They were very critical and helpful for improving our work. We have deleted this paragraph in part of introduction, and critically reviewed and revised the part of the materials and methods.
5. (Methods and materials): Need to be revised critically. Please note to comments below:What is the type of study? What was the inclusion criteria of study? When was blood sampling done? Where was blood sampling done?EDTA? Was flow cytometry done for first group (OC group) not for BOT and control groups?
Response: Thank you for these comments. They were very critical and helpful for improving our work. Our study belongs to original research articles, and inclusion criteria of study, sampling time, sample type and group information, etc are all added to the part of materials and methods. We have critically reviewed and revised the part of the materials and methods, and the revised content is shown in the yellow part of the manuscript (lines 96-102, page 3; lines 107-112, page 4).
6. (Statistical analysis): Please revives statistical analysis section. “Qualitative variables were reported as percentages and quantitative variables were reported with mean and standard deviation”.The differences between groups were compared using an independent-samples t test?, one-way ANOVA ?and Mann-Whitney U tests test. For which groups mentioned test were used?
Response: Thank you for these comments. They were very critical and helpful for improving our work. We have detailed the statistical analysis between different groups in the yellow part of the manuscript (lines 137-146, page 4).
7. (Results): The proportion of CD4+CD25+ Treg cells was higher in OC patients than in healthy controls (8.02 ± 4.24 vs 5.15 ± 3.95, P<0.05). How about the BOT groups? Result section is difficult to be followed. Please draw a table and post some of information and data of three groups (like proportion of Treg cells (Foxp3+, CD25+Foxp3+) in CD4+ T cells, …)
Response: Thank you for these comments. They were very critical and helpful for improving our work. We have shown the results of this part in the form of figure1(Table2). According to your suggestion, we have added the tabular form of comparison results of different T cell subsets among the three groups OC, BOT and HC, and the revised content is shown in the yellow part of the manuscript (Table2, page 7).
We would like to thank the reviewers and editors again for your positive and helpful comments and for the opportunity to revise our manuscript by incorporating these recommended changes.
We are looking forward to your positive decision!
Sincerely,
Fang Wang, MD., PhD.
Department of Laboratory Medicine
the First Affiliated Hospital of Nanjing Medical University
Nanjing, China
Phone: +86-86862814
Email: [email protected]
Round 2
Reviewer 1 Report
I accept the changes made. The manuscript is still written in a very complicated way. Perhaps for a certain group of readers, it will be interesting and useful.
Author Response
Dear Ms.Winifred Zhao and Reviewers:Thank you very much for your kindly reply on our manuscript entitled “Circulating CD4+ Treg, CD8+ Treg and CD3+ γδT cells subpopulations in ovarian cancer” (medicina-2057434). We gratefully appreciate for your very insightful comments, these comments are all valuable and very helpful for revising and improving our paper, as well as the important guiding significance to our research. We checked the references cited in the manuscript, optimized all the tables again, and confirmed the correctness of all our statistical methods. We made revisions by using a yellow font color or a yellow background to flag revised positions which were changed in the manuscript. All pages and lines have been numbered in the revised version. We sincerely hope that our revised manuscript could be approved. In addition, a point-by-point reply to the reviewers’ concerns is included below.
Reviewers’ Comments:
Referee #1 (Remarks to the Author):
I accept the changes made. The manuscript is still written in a very complicated way. Perhaps for a certain group of readers, it will be interesting and useful.
Response: Thank you for these comments. We will continue to practice improving our paper writing ability and skills. In addition, we will make further efforts to explore the characteristics of Tregs in OC microenvironment and provide clinical basis for future treatment based on Tregs. Thank you again for taking your precious time to review our manuscript.
We would like to thank the reviewers and editors again for your positive and helpful comments and for the opportunity to revise our manuscript by incorporating these recommended changes.
We are looking forward to your positive decision!
Sincerely,
Fang Wang, MD., PhD.
Department of Laboratory Medicine
the First Affiliated Hospital of Nanjing Medical University
Nanjing, China
Phone: +86-86862814
Email: [email protected]
Reviewer 3 Report
Dear authors many thanks for the revisions.
Please revise:
1. all abbreviation words need to be omitted from the title of tables. Please write full of the words at the titles.
2. all abbreviations included in the tables should be defined on the legend of the tables.
3. Table 2. is not clear, there are some signs that there is no explanation for them. Please clarify those signs and add a column and compare the variables of the three groups statistically.
4. Please at the legend of all tables to determine which statistical tests were used for variables. For example in table 3, for "age with p=0.015" which test was used?
Author Response
Dear Ms.Winifred Zhao and Reviewers:
Thank you very much for your kindly reply on our manuscript entitled “Circulating CD4+ Treg, CD8+ Treg and CD3+ γδT cells subpopulations in ovarian cancer” (medicina-2057434). We gratefully appreciate for your very insightful comments, these comments are all valuable and very helpful for revising and improving our paper, as well as the important guiding significance to our research. We checked the references cited in the manuscript, optimized all the tables again, and confirmed the correctness of all our statistical methods. We made revisions by using a yellow font color or a yellow background to flag revised positions which were changed in the manuscript. All pages and lines have been numbered in the revised version. We sincerely hope that our revised manuscript could be approved. In addition, a point-by-point reply to the reviewers’ concerns is included below.
Reviewers’ Comments:
Referee #3 (Remarks to the Author):
- all abbreviation words need to be omitted from the title of tables. Please write full of the words at the titles.
Response: Thank you for these comments. We have replaced the full of the words with abbreviations in all table titles in the manuscript to make it easier for readers to read, and the revised content is shown in the yellow part of the manuscript (page 7, 9-11, 14).
- all abbreviations included in the tables should be defined on the legend of the tables.
Response: Thank you for these comments. We have defined all abbreviations on the legend of the tables in manuscript, and the revised content is shown in the yellow part of the manuscript (page 7, 9-11, 14).
- Table 2. is not clear, there are some signs that there is no explanation for them. Please clarify those signs and add a column and compare the variables of the three groups statistically.
Response: Thank you for these comments. We have explained all signs and added a column that compared the variables of the three groups statistically. (Line 196, page 7). 4. Please at the legend of all tables to determine which statistical tests were used for variables. For example, in table 3, for "age with p=0.015" which test was used?
Response: Thank you for these comments. We have determined the statistical tests that used for variables at the legend of all tables. For example, independent-sample t test was used for "age with p=0.015" in table 3. Moreover, the statistical tests that used in each table was listed below:
Tables |
Statistical tests |
||
independent-sample t test |
One way ANOVA |
Mann-Whitney U |
|
Table 2 |
□YES □NO |
YES □NO |
□YES □NO |
Table 3 |
YES □NO |
YES □NO |
YES □NO |
Table 4 |
YES □NO |
YES □NO |
YES □NO |
Table 5 |
YES □NO |
YES □NO |
□YES □NO |
Table 6 |
YES □NO |
□YES □NO |
YES □NO |
“”indicate the test was used in this table.
We would like to thank the reviewers and editors again for your positive and helpful comments and for the opportunity to revise our manuscript by incorporating these recommended changes.
We are looking forward to your positive decision!
Sincerely,
Fang Wang, MD., PhD.
Department of Laboratory Medicine
the First Affiliated Hospital of Nanjing Medical University
Nanjing, China
Phone: +86-86862814
Email: [email protected]
